# Evaluation of flow control using PID versus fuzzy logic in an electropneumatic circuit for pulmonary ventilation applications

Lina González[1,☉], Issa Griffith[1,☉], Alfredo Lescher[1,☉], Jay Molino[1,2*], Asdrúal Rojas[1], Damián Quijano[1]

**1** Centro I+D+i de Biotecnología, Energías Verdes y Cambio Climático (BEVCC), Laboratorio de Investigación Experimental de Bioseñales, Biomedical Engineering, Faculty of Biosciences and Public Health, Universidad Especializada de las Americas (UDELAS), Albrook, Paseo de La Iguana, Panama, **2** Sistema Nacional de Investigación (SNI), SENACYT, Panama City, Republic of Panama

☉ These authors contributed equally to this work.
* jay.molino@udelas.ac.pa

## Abstract

High-tech mechanical ventilators are engineered to deliver precise and consistent airflow, which is critical for effective respiratory therapy. This study evaluates flow control performance in a custom-built electro-pneumatic ventilator prototype, comparing Proportional-Integral-Derivative (PID) control with Fuzzy Logic Control (FLC) through real-time experiments on a test-lung platform to assess accuracy and adaptability under dynamic conditions. A laboratory based experimental study was conducted under laboratory conditions, using a test lung simulator and real-time flow data acquisition. The analysis included time-domain performance metrics and statistical validation through Bland–Altman analysis. Results indicate that both controllers meet the accuracy thresholds expected in commercial systems. However, the fuzzy logic controller exhibited narrower limits of agreement and lower standard deviation, indicating greater consistency. While PID control responded faster, with a settling time between 0.32 and 0.43 seconds, FLC achieved superior performance in high-demand scenarios, delivering an entire volume of 900 mL. Stability analysis using the Jury Test and Nyquist criteria confirmed that both systems are dynamically stable. Notably, the FLC curve in the Nyquist plot remained farther from the critical point (−1, 0j), indicating enhanced robustness against disturbances. These findings suggest that FLC may offer a reliable alternative to PID in nonlinear ventilation scenarios, particularly in resource-constrained environments seeking technological autonomy.

## Introduction

Mechanical ventilators are essential for assisting or replacing spontaneous breathing by regulating airflow, pressure, volume, and timing [1]. Precise control is crucial to

---

**Data availability statement:** All relevant data are within the article and its Supporting Information files.

**Funding:** This work was supported by the Secretaría Nacional de Ciencia, Tecnología e Innovación (SENACYT), Panama, under grant number APY-NI2022-13 awarded to Lina González. The funders had no role in study design, data collection and analysis, decision to publish, or preparation of the manuscript.

**Competing interests:** The authors have declared that no competing interests exist.

prevent ventilation errors or barotrauma. Modern ventilators rely on advanced control systems, with Proportional-Integral-Derivative (PID) and Fuzzy Logic controllers being two widely studied approaches [2]. However, a research gap remains in understanding their comparative performance under realistic conditions, critical for improving ventilator design, especially during ventilator shortages in global health crises [3].

Mechanical ventilation has evolved from manual devices to microprocessor-driven systems with feedback controls [4]. PID control is widely used due to its simplicity and reliable performance in regulating airflow and pressure [5]. Valve positions are continuously adjusted using tuned gains based on measured vs. desired values. While effective in stable conditions, PID struggles with nonlinearities and patient variability, leading to overshoot or oscillations when conditions change [6]. External disturbances like coughing or circuit leaks further challenge PID, sometimes causing delayed responses or excessive pressure fluctuations [7].

Despite its fast response and ease of implementation, PID's limitations necessitate alternative approaches like Fuzzy Logic Control (FLC), which may offer superior adaptability in nonlinear environments [8]. However, a direct experimental comparison of PID and FLC in ventilator control is lacking. Our study addresses this by evaluating both side by side in an electro-pneumatic ventilator prototype, providing evidence for optimal control strategies in future ventilator designs [9].

Beyond academic interest, this comparison has practical significance. The COVID-19 pandemic exposed ventilator shortages, driving demand for cost-effective and adaptive control systems [10]. If FLC outperforms PID, it could revolutionize ventilator technology, especially in resource-limited settings. Conversely, if PID proves sufficient with proper tuning, it reinforces its continued use. Understanding these comparative behaviors will help guide future ventilator designs [11].

Controlling flow in an electro-pneumatic ventilator is challenging due to nonlinear system dynamics and component characteristics [12]. The prototype in this study includes an ESP32 microcontroller, pneumatic valves (a proportional valve for fine control and a solenoid valve for rapid actuation), pressure and flow sensors, and a test lung interface. The nonlinearity arises from multiple factors: valve flow depends on input signals and pressure differences, lung inflation alters compliance and resistance in real-time, and sensors introduce measurement lags or inaccuracies [13]. Additionally, actuator hysteresis and switching delays affect ventilator response, making stability across different conditions difficult. Controllers must adapt to these variations to ensure reliable operation [14].

PID controllers, being linear, do not inherently adjust to changing system dynamics. They require manual tuning to balance response speed and stability [15]. Engineers often use heuristic methods like Ziegler–Nichols tuning, which determines gains based on system oscillations [16]. However, Ziegler–Nichols tuning tends to produce aggressive gains, often leading to overshoot in sensitive applications like ventilation [17]. As a result, additional trial-and-error tuning is needed to refine PID performance, making the process time-consuming and requiring expertise [18]. PID gains remain a compromise even when tuned, potentially underperforming in pediatric patients or cases of increased airway resistance due to secretions [19].

To address these issues, adaptive PID strategies—such as Tyreus-Luyben tuning or auto-tuning algorithms—have been investigated to improve response across different conditions [20]. Anitha et al. compared multiple PID tuning methods and found that structured tuning improved performance, though adaptation remained a limitation [21]. Thus, while PID control is simple and effective, its dependency on static tuning limits its robustness in highly variable ventilatory conditions [22].

To mitigate fixed-gain PID limitations, researchers have explored intelligent and adaptive control techniques [23]. Model-based adaptive control, for example, dynamically adjusts control parameters based on lung mechanics. Hünnekens et al. developed a variable-gain controller that adapts to real-time lung conditions, enhancing stability across changing scenarios [8]. Similarly, clinical ventilator modes like Adaptive Support Ventilation (ASV) use algorithm-driven adjustments to optimize tidal volume and respiratory rate. Though not AI-based, these systems incorporate advanced rule-based logic to automate ventilator settings, reducing clinician workload and improving patient-ventilator synchronization [24].

AI and machine learning transform ventilator control through neural networks, fuzzy inference systems, and reinforcement learning [23]. Adaptive neuro-fuzzy inference systems (ANFIS) combine neural networks with fuzzy logic for self-tuning controllers [14]. Živčák et al. implemented ANFIS in a pneumatic ventilator, improving performance over fixed controllers [15]. Machine learning models now predict patient-specific parameters, enabling real-time ventilation adjustments [16]. Xiong et al. reviewed AI in ventilation, noting a shift from decision support (e.g., weaning prediction) to closed-loop AI controllers that autonomously adjust ventilator settings [23]. While AI controllers adapt dynamically, many remain experimental due to complexity and validation challenges [23].

Fuzzy Logic Control (FLC) stands out among intelligent control methods for its transparency and rule-based adaptability [17]. Unlike neural networks, FLC requires no training data, relying instead on linguistic rules (e.g., "if pressure error is high, close the valve slightly") [18]. FLC handles nonlinearities well, making it ideal for highly variable respiratory mechanics [19]. El Adawy et al. used fuzzy control for oxygen therapy, achieving stable regulation without lung modeling [15]. Similarly, Ospino Castro et al. developed a fuzzy-based oxygen regulator, improving blood oxygen stabilization [16]. However, most past applications lacked direct comparisons to PID control, limiting conclusions about FLC's superiority. This study addresses that gap by directly comparing FLC and PID under identical test conditions [20].

Fuzzy controllers operate using four steps: (1) fuzzification (converting sensor inputs into fuzzy values), (2) a rule base (expert-defined "if-then" statements), (3) inference (evaluating rules), and (4) defuzzification (converting fuzzy outputs into precise control actions) [21]. Zadeh's 1965 work laid the foundation for fuzzy logic in control systems [22].

In our electro-pneumatic ventilator, FLC regulates inspiratory flow based on flow error and its rate of change, a widely used approach in dynamic systems [23]. We chose triangular membership functions for simplicity and smooth control, as seen in inverted pendulum and ventilator control studies [12]. Studies suggest that 3–5 membership functions per input perform well while maintaining interpretability [18].

Our fuzzy rule base mimics an experienced respiratory therapist, adjusting valve commands based on error trends [19]. For example, "IF flow error is large positive AND increasing, THEN open valve rapidly." We constructed 25 rules (5×5 matrix) from PID behaviors, refining them through observation—similar to Chao et al.'s work, which mapped PID behavior into fuzzy regulations [18].

Justification of Membership Functions and Rules: We prioritized symmetric triangular functions for balanced sensitivity and stability, avoiding overly complex models that risk overfitting [20]. Recent AI techniques optimize fuzzy rule weights and membership functions [21]. Allagui et al. developed an AI-assisted fuzzy-PID hybrid, improving robotic control and hinting at future machine learning-enhanced fuzzy tuning [22]. Preliminary tests showed stable fuzzy control, with quick settling and no oscillations, reinforcing its potential before direct PID comparison [24].

This study bridges classical and intelligent control for mechanical ventilation by directly comparing PID and Fuzzy Logic controllers under identical conditions [10]. While prior research has explored these methods separately or in simulations, real-world comparisons remain scarce [21]. We provide a fair, experimental evaluation by implementing both controllers on the same electropneumatic ventilator prototype and testing their response to setpoint changes, lung mechanics

variability, and disturbances [23]. Key performance metrics include rise time, settling time, overshoot, steady-state error, and stability margins, ensuring alignment with standard control benchmarks [24].

Beyond determining which controller performs better, we investigate why, for instance, FLC's nonlinearity improves performance in certain conditions or whether PID's simplicity enables faster reaction to sudden changes [19]? Ultimately, this study fills a knowledge gap in ventilator control, guiding the future development of robust, practical, and precise respiratory support systems through traditional PID or intelligent Fuzzy Logic control [22].

## Methods and experimental setup

### Electropneumatic ventilator prototype: Design and components

The electro-pneumatic ventilator prototype was developed to regulate airflow and pressure with Proportional-Integral-Derivative (PID) Control and Fuzzy Logic Control (FLC). The system integrates an ESP32 microcontroller, a pneumatic actuation system, pressure and flow sensors, and a test lung simulator to evaluate its performance under controlled conditions. Validation was performed against ISO 80601-2-12:2021 standards to ensure accuracy and reliability.

The mechanical system consists of a two-valve structure designed for precise inspiratory and expiratory airflow control. The proportional valve (Series AP, Camozzi) enables continuous fine-tuned regulation of inspiratory flow, adjusting dynamically based on control inputs. The solenoid valve (Series CFB, Camozzi) is responsible for expiratory airflow modulation, operating in a binary manner to facilitate rapid switching between phases. To ensure stable operation, the prototype incorporates a medical-grade air compressor (Aridyne 3500, Allied Medical LLC) that supplies the necessary pressurized air, regulated by a micro pressure regulator (Series M, Camozzi) to maintain consistent airflow delivery. The system also includes a flow sensor (Sensirion SFM3000), which provides real-time flow measurements as feedback for control algorithms. The ventilator output is connected to a test lung simulator (ACCU LUNG, Fluke), which replicates patient respiratory mechanics by allowing adjustable compliance and resistance settings. The main components are details in Table 1. Fig 1 depicts the circuit schematic (left panel) and the assembled electropneumatic ventilator prototype (right panel), with labeled components.

### System nonlinearities and dynamic characteristics

The electropneumatic ventilator system exhibits nonlinear behavior due to mechanical constraints, sensor feedback delays, and dynamic lung compliance changes. These nonlinearities introduce control challenges, requiring robust tuning strategies. One of the primary sources of nonlinearity arises from the proportional valve, which demonstrates hysteresis effects and a nonlinear relationship between input voltage and flow rate. Meanwhile, the solenoid valve creates abrupt airflow transitions, affecting cycle timing and flow stability.

Another key source of variability stems from lung compliance and resistance fluctuations, as the test lung simulates different pulmonary conditions, altering airway resistance and dynamic lung mechanics in real time. The flow sensor (Sensirion SFM3000) introduces additional measurement lags and noise, necessitating real-time filtering techniques to

**Table 1. Key components and specifications.**

| Component | Description | Specification |
|---|---|---|
| Microcontroller | ESP32 dual-core processor | 240 MHz, Wi-Fi/BLE |
| Proportional Valve | Fine-tuned flow regulation | 0–10V input, response <5ms |
| Solenoid Valve | Inspiratory/expiratory actuation | 24V DC, switching time <10ms |
| Pressure Regulator | Ensures stable input pressure | Adjustable: 0–10 bar |
| Flow Sensor | Measures dynamic respiratory airflow | ±5% accuracy |
| Air Compressor | Medical-grade compressed air supply | Max 100 psi, 50L/min output |
| Test Lung | Simulates lung mechanics | Adjustable compliance settings |

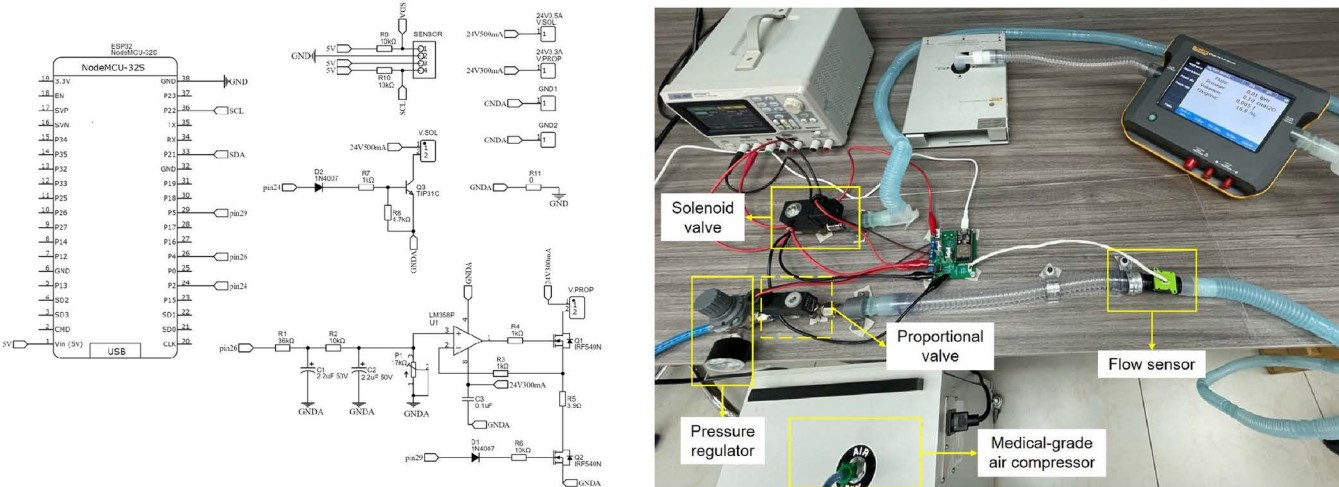

**Fig 1. (Left panel) Schematic of the electronic circuit; (Right panel) Electropneumatic Unit.**

maintain accuracy. Furthermore, actuator response delays due to valve hysteresis and finite switching times introduce further complexities, impacting the system's ability to respond swiftly to patient variability.

Given these dynamic behaviors, the control methodology selection (PID vs. Fuzzy Logic) is crucial, as it determines the system's ability to adapt to nonlinear variations and maintain stable operation under real-world conditions.

### PID controller design and tuning

The PID controller is responsible for precisely regulating valve positions to achieve the desired inspiratory flow and pressure levels. To determine the optimal controller gains, the Ziegler-Nichols tuning method was initially applied to estimate the critical gain (Kc) and oscillation period (Tu), followed by experimental fine-tuning to enhance system performance. The final PID tuning values are summarized in Table 2.

Tuning adjustments were carefully optimized to minimize steady-state error, ensuring precise airflow delivery while reducing overshoot to prevent barotrauma risks. Rise and settling times were optimized to enhance responsiveness to sudden respiratory demand changes. An adaptive tuning refinement was implemented to improve adaptability, allowing for dynamic adjustments in response to lung compliance variability and sensor drift.

### Fuzzy Logic Controller (FLC) design

To address the limitations of PID in handling nonlinear respiratory mechanics, a Fuzzy Logic Controller (FLC) was designed to dynamically adjust inspiratory flow based on real-time variations in lung mechanics (Table 3). Unlike PID, which relies on fixed gain parameters, FLC mimics human decision-making by continuously evaluating airflow deviations and rate of change, adjusting the control response accordingly. The fuzzy control system consists of four key components:

**Table 2. PID tuning parameters.**

| Gain Parameter | Value |
| --- | --- |
| Proportional Gain (Kp) | 5.7 |
| Integral Gain (Ki) | 87.3 |
| Derivative Gain (Kd) | 0.05 |

**Table 3. Fuzzy logic control rules.**

| Flow Error | Error Rate | Control Action |
|---|---|---|
| Large Positive | Increasing | Open valve rapidly |
| Small Positive | Constant | Maintain current opening |
| Zero | Any | No change |
| Small Negative | Decreasing | Reduce valve opening |
| Large Negative | Rapid Decreasing | Close valve quickly |

1. *Fuzzification:* Converts flow error and error rate into linguistic variables.

2. *Rule Base:* A 5×5 matrix defines 25 control rules, mapping error trends to control actions.

3. *Inference Mechanism:* Uses Mamdani-type fuzzy inference to determine the degree of rule activation.

4. *Defuzzification:* Translates fuzzy outputs into precise valve control signals.

The triangular membership functions were carefully selected to provide efficient computation while ensuring smooth transitions between control states. The rule base was optimized through experimental testing, balancing interpretability with adaptive control precision.

## Experimental validation

A controlled experimental setup was implemented using the test lung simulator to evaluate PID vs. Fuzzy Logic performance. The validation process included three key tests:

1. *Step Response Tests:* Evaluated rise time, settling time, and overshoot in response to sudden setpoint changes.

2. *Disturbance Rejection Tests:* Assessed controller robustness under simulated disturbances, including patient coughs and circuit leaks.

3. *Bland-Altman Analysis*: Verifying accuracy and precision compared ventilator performance against a commercial reference ventilator.

The results demonstrated quantifiable differences in control performance, providing empirical evidence for the effectiveness of PID and FLC strategies under realistic ventilatory conditions.

The novelty of our approach lies in the direct empirical comparison of PID and Fuzzy Logic Control under identical hardware conditions in a real-time electro-pneumatic ventilation platform. Unlike previous studies that rely on simulation-only environments, this prototype integrates low-cost components and validates controller performance under dynamically varied respiratory compliance and disturbances. Using an ESP32-based control system with modular code and open-source libraries also offers a replicable framework for cost-effective ventilator design in resource-limited settings.

## Results

The analysis covers several performance metrics, including settling time, overshoot, volume accuracy, stability margin, and transient oscillations.

### Conformity evaluation

The conformity evaluation was conducted using the electro-pneumatic test bench, comparing its measurements with the Fluke VT650 flow analyzer under both PID and Fuzzy Logic control. A Bland-Altman analysis revealed mean differences of −0.122 L/min for PID and −0.193 L/min for Fuzzy Logic control (Table 4).

**Table 4. Bland-Altman analysis results.**

| Control | Parameter | Value | S.E. | Inferior | Superior |
|---|---|---|---|---|---|
| PID | Mean Difference | −0.122 | 0.059 | −0.238 | −0.005 |
| | Lower Limit | −1.269 | 0.100 | −1.468 | −1.070 |
| | Upper Limit | 1.025 | 0.100 | 0.826 | 1.225 |
| Fuzzy Logic | Mean Difference | −0.193 | 0.055 | −0.302 | −0.084 |
| | Lower Limit | −1.263 | 0.094 | −1.449 | −1.076 |
| | Upper Limit | 0.876 | 0.049 | 0.690 | 1.063 |

The standard error of the mean (S.E.) was 0.059 for PID and 0.055 for Fuzzy Logic control. The limits of variability ranged from −1.2688 to 1.0255 L/min for PID and from −1.2627 to 0.8765 L/min for Fuzzy Logic control.

The mean differences observed for both controls compared to the VT650 analyzer were not significant, aligning with the precision of recognized commercial equipment [25–27]. The experimental readings were precise, suggesting robustness in the control system's ability to replicate measurements under controlled conditions.

The Bland-Altman graph (Fig 2) visualized these differences against the average measurements of the control and analyzer. The distribution of disagreements showed no clear pattern of proportional bias, indicating that discrepancies did not systematically increase or decrease with the measured flow levels.

The Bland-Altman analysis for each volume test with both controllers demonstrated notable similarity and consistency with previous results. Both controls fell within the precision limits of recognized commercial equipment. Still, the Fuzzy Logic control showed better standard deviation and limits of differences, indicating higher reliability in reproducing consistent and predictable flow results, which is crucial in managing critical patients where ventilator stability is essential.

The visual inspection of the Bland-Altman graphs (Fig 2) showed that most differences for Fuzzy Logic control were closely clustered around the mean line and within the limits of agreement, indicating higher precision and consistency. The mean difference line for Fuzzy Logic control was closer to zero, suggesting a less average underestimation of flow compared to the Fluke analyzer. Additionally, the differences beyond the limits were less dispersed, indicating fewer and less pronounced extreme errors than PID control.

## Repeatability

The Fisher-Snedecor statistical test was applied to evaluate repeatability, calculating the required sample size using the formula for a finite sample as shown in Equation (1).

$$n = \frac{N \cdot Z_\alpha^2 \cdot p \cdot q}{e^2 \cdot (N-1) + Z_\alpha^2 \cdot p \cdot q}$$

(1)

The sample size calculation is based on several key parameters. Here, n represents the sample size, while N denotes the population size. Here, Z represents the critical value of the standard normal distribution corresponding to the desired confidence level and e is the maximum acceptable error. The probability of success is indicated by p, with q representing the probability of failure, defined as $(1-p)$. For the electro-pneumatic test bench, with a population of 4000 records, a 95% confidence level (Z = 1.96Z = 1.96Z = 1.96) and a 5% maximum acceptable error were used, resulting in a calculated sample size of 71.70. Similarly, for the analyzer, which had a population of 2000 records, the sample size was determined to be 70.40.

Stable state data were selected for the Fisher test with a significance level of 0.05, and they were applied to each controller and volume test. Results are shown in Tables 5 and 6.

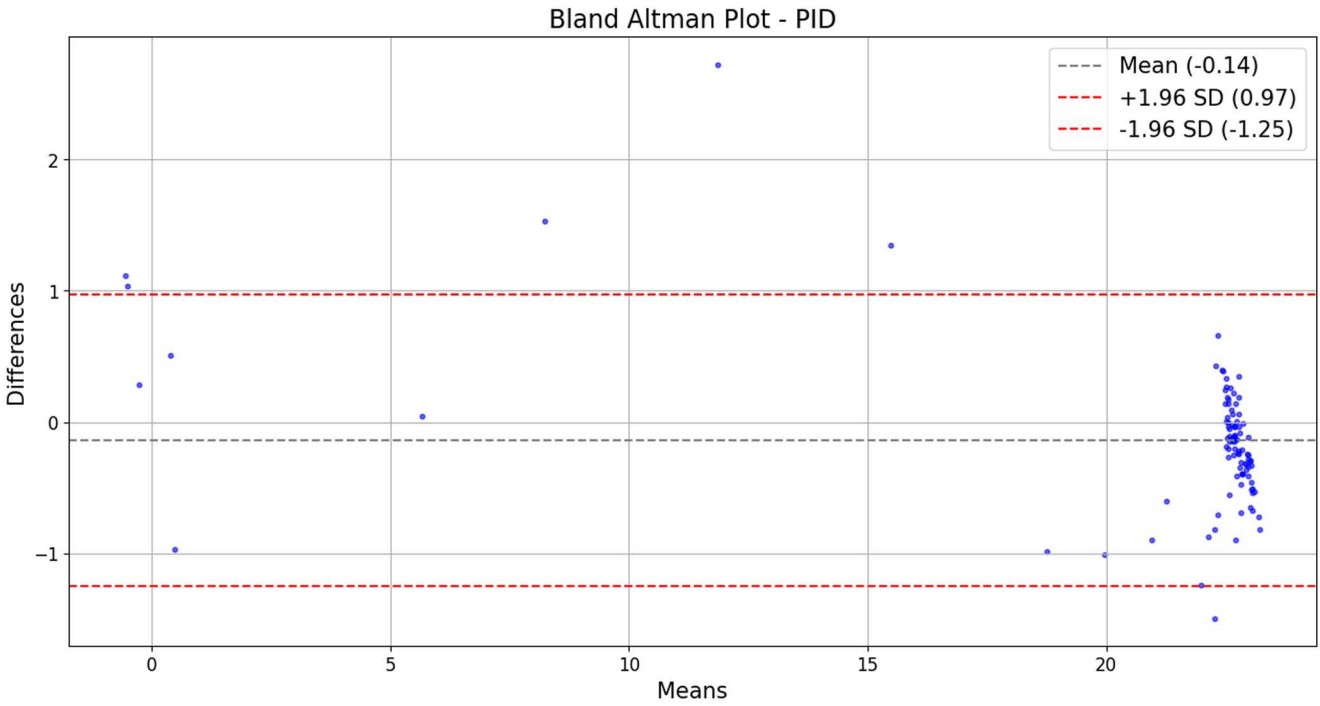

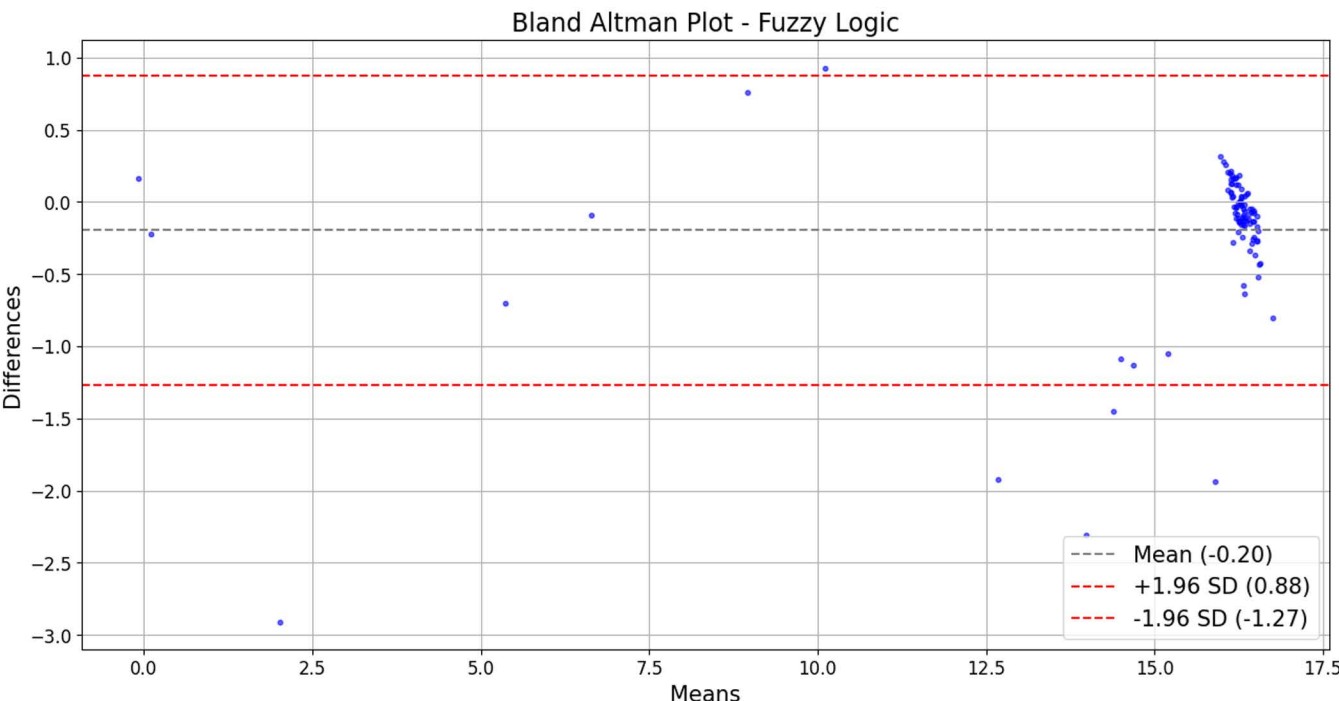

**Fig 2. Bland-Altman analysis.** (Top Panel) PID Control; (Bottom Panel) Fuzzy Logic Control.

The F-test results compare the means' hypotheses:

• *Null Hypothesis (H0): No statistically significant difference between group variances.*

• *Alternative Hypothesis (H1): Statistically significant difference between group variances.*

Hypotheses are accepted or rejected by comparing F values to the critical F value. If F < critical F, the null hypothesis is rejected. Additionally, P values were compared to the significance level (P ≤ 0.05) for validation.

Tables 5 and 6 analysis shows consistent results supporting the null hypothesis. F values for the PID controller (500 ml, 700 ml, 900 ml) were 1.191, 1.280, and 1.033, all below critical F values. For the Fuzzy Logic controller, F values (500 ml, 700 ml, 900 ml) were 1.093, 1.469, and 1.113; all values are below critical F values. These results are enough to declare statistical significance for both controllers.

Besides, P values for Fuzzy Logic controller tests were consistently above 0.05, reinforcing the null hypothesis. These results indicate robust control systems maintaining volume delivery homogeneity under test conditions, suggesting high repeatability.

### Settling time

The settling time refers to the period required for the system output to remain within a specified range between 2% and 5% of the desired final steady-state value [28]. For this analysis, we define the settling time as when the output remains within ±2% of the final value.

The PID controller showed exceptional settling time performance, with average times of 0.43s for 500 mL, decreasing to 0.35s for 700 mL and 0.32s for 900 mL. It also indicates a fast response, which is crucial in clinical situations requiring precise adjustments.

In contrast, the Fuzzy Logic controller presented longer times, with 0.77s for 500 mL, improving to 0.43s for 700 mL and 0.58s for 900 mL.

### Stability

**Jury analysis.** Stability was evaluated using Jury analysis of the transfer functions for each control, obtained with Matlab's System Identification Toolbox. The resulting transfer function for the Fuzzy Logic control is presented in Equation (2). Equation (3) shows the transfer function obtained for the PID control. In both equations, the variable z denotes the complex frequency domain variable used in Z-transform analysis for discrete-time systems.

**Table 5. Fisher test for PID controller.**

| Volume (mL) | Test Bench Variance | Analyzer Variance | Test Bench Std Dev | Analyzer Std Dev | F | P (F<=f) | Critical F |
|---|---|---|---|---|---|---|---|
| 500 | 0.003 | 0.002 | 0.054 | 0.050 | 1.191 | 0.233 | 1.487 |
| 700 | 0.003 | 0.004 | 0.056 | 0.063 | 1.280 | 0.152 | 1.485 |
| 900 | 0.004 | 0.004 | 0.064 | 0.065 | 1.033 | 0.446 | 1.485 |

**Table 6. Fisher test for Fuzzy Logic Controller.**

| Volume (mL) | Test Bench Variance | Analyzer Variance | Test Bench Std Dev | Analyzer Std Dev | F | P (F<=f) | Critical F |
|---|---|---|---|---|---|---|---|
| 500 | 0.002 | 0.002 | 0.050 | 0.048 | 1.093 | 0.355 | 1.487 |
| 700 | 0.003 | 0.004 | 0.052 | 0.063 | 1.469 | 0.055 | 1.485 |
| 900 | 0.003 | 0.004 | 0.056 | 0.059 | 1.113 | 0.327 | 1.485 |

$$G_{FUZZY}(z) = \frac{0.5055 - 0.4212z^{-1} + 0.01324z^{-2} + 0.02104z^{-3} + 0.05695z^{-4}}{1 - 0.2329z^{-1} + 0.1049z^{-2} - 0.1606z^{-3} + 0.1928z^{-4}} \tag{2}$$

$$G_{PID}(z) = \frac{0.0141 + 0.06238z^{-1} - 0.07216z^{-2} + 0.2293z^{-3} - 0.07586z^{-4}}{1 - 0.6273z^{-1} + 0.442z^{-2} - 0.3457z^{-3} + 0.1437z^{-4}} \tag{3}$$

The results indicate that both the PID and Fuzzy Logic controllers meet the four conditions established by the Jury Test [29]. Once all the stability conditions are satisfied, it is concluded that the given characteristic equations are stable. Furthermore, when graphically verifying the system's poles, they all appear within the unit circle of the Z plane. This confirms that the stability criteria for the discrete mode controllers of the presented design are met.

**Nyquist analysis.** Nyquist plots (Fig 3) show the relationship between the real and imaginary parts of the system's frequency response. Both controls do not encircle the critical point (−1, 0j), indicating stability [29].

The fuzzy logic control plot (Fig 3- Top Panel) is farther from the origin and the critical point than PID control (Fig 3-Bottom Panel), indicating better stability and robustness against variations. The multiple loops in the PID control plot suggest pronounced frequency response peaks, indicating sensitivity at specific frequencies. The fuzzy logic control plot indicates smoother, less resonant behavior, which is beneficial for robustness against operational variations and disturbances.

Gain and phase margins were also evaluated. Fuzzy logic control showed infinite gain and phase margins, indicating tolerance to any gain or phase shift before instability. PID control had a gain margin of 15.58 dB, indicating some tolerance to gain before instability.

## Discussion

The experimental evaluation of the fuzzy logic and PID controllers for the electro-pneumatic ventilator system demonstrates several key findings that highlight the advantages of the fuzzy logic controller. The results show that both systems are within the accuracy margins of commercial equipment. Still, Fuzzy Logic, with lower variability in measurements according to Bland-Altman analysis, appears more accurate and consistent. Regarding repeatability, when comparing the results of both controllers with the established target flows, it is perceived that both Fuzzy Logic and PID have a high capacity to reach the target flows with minimal deviations.

Although PID achieves faster settling times, Fuzzy Logic fits better at higher volumes, showing better adaptability, which is crucial in dynamic clinical environments. Stability analyses, including Nyquist plots and Jury analysis, indicate that both systems are stable. However, the Fuzzy Logic controller exhibits a smoother and less resonant frequency response, suggesting greater robustness to variations, which is beneficial in complex environments.

Further development and refinement of fuzzy logic algorithms are essential. Optimizing parameters and incorporating machine learning techniques could enhance the adaptability and precision of the controller, allowing it to adjust dynamically to real-time data. Controlling flow in neonatal patients presents unique challenges due to the low volumes and high frequencies involved. Future studies should focus on this scenario to ensure that the fuzzy logic controller can handle these demanding conditions effectively. Both controllers' algorithms should be optimized, especially for high-flow situations where the PID controller has shown better responsiveness. Enhancing the fuzzy logic controller's performance in these conditions could further establish its superiority.

## Conclusion

This study has demonstrated the superior performance of Fuzzy Logic control over traditional PID control in the context of mechanical ventilation systems, particularly in terms of stability and robustness. The data analysis clearly shows that Fuzzy Logic control provides enhanced adaptability and precision, making it a more effective solution for managing mechanical ventilation's complex and dynamic requirements. These findings are especially relevant in scenarios where patient-specific adjustments and nonlinearities in system behavior are critical. For practitioners, the results of this study

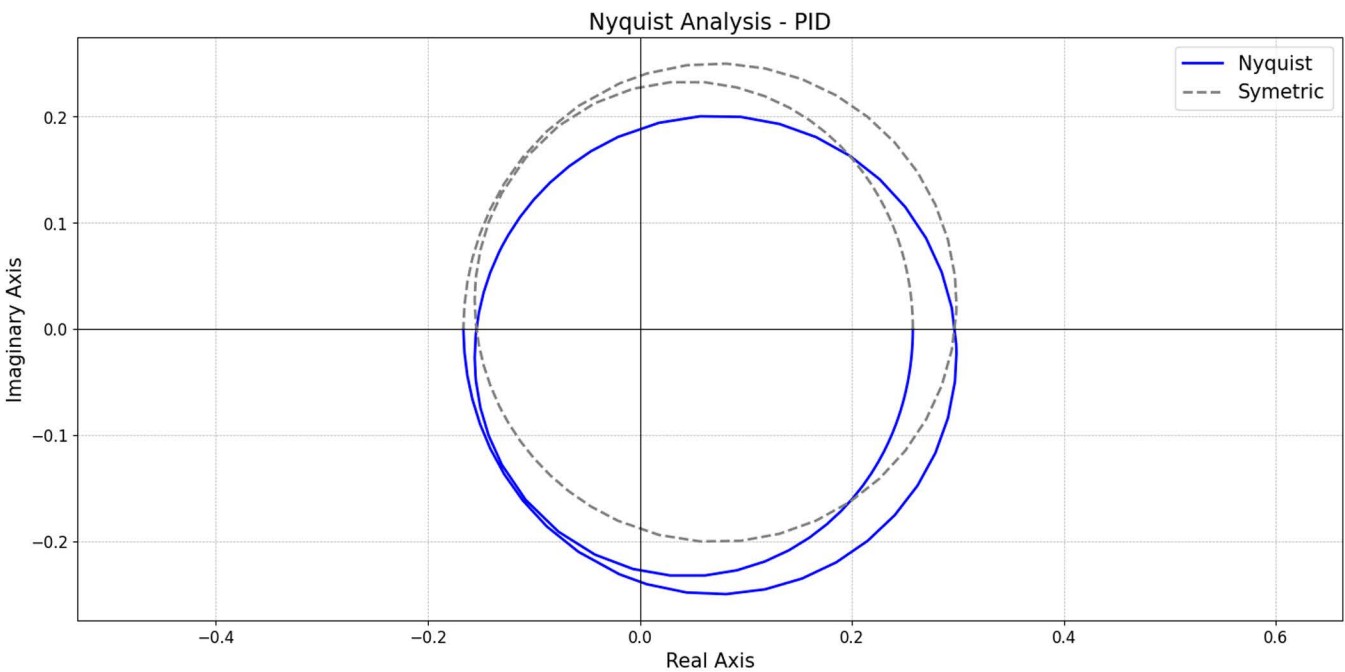

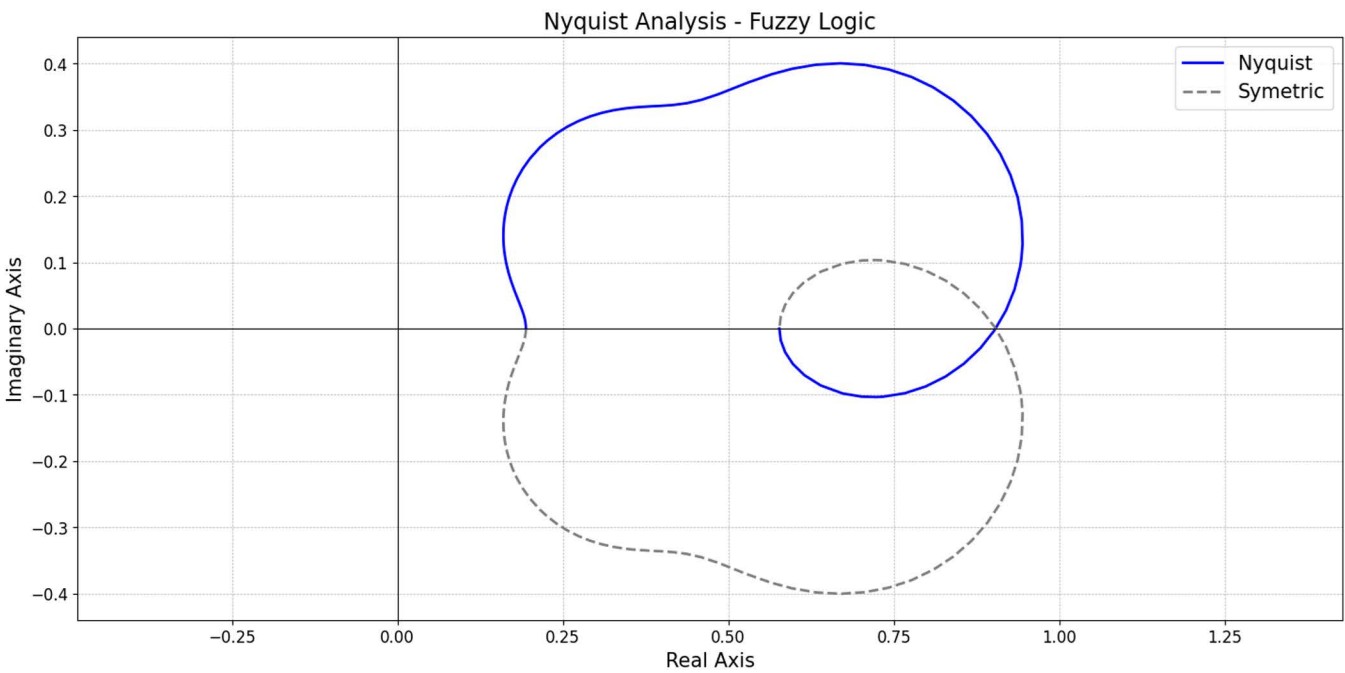

**Fig 3. Nyquist diagrams.** (Top Panel) PID Control; (Bottom Panel) Fuzzy Logic Control.

suggest that integrating Fuzzy Logic control into mechanical ventilation systems could significantly improve patient outcomes by offering more consistent and reliable respiratory support. Practitioners should consider the adoption of Fuzzy Logic controllers in environments where precise control of ventilation parameters is essential, such as in intensive care

units or during critical surgical procedures. For policymakers, the research underscores the importance of supporting the development and implementation of advanced control technologies in medical devices. Policies encouraging the integration of Fuzzy Logic control in medical equipment could improve healthcare delivery, particularly in emerging and developing economies where access to high-quality medical technology is often limited. Additionally, establishing standards and guidelines for using Fuzzy Logic control in medical devices could further enhance patient safety and care quality.

## Author contributions

**Conceptualization:** Lina González, Issa Griffith, Alfredo Lescher, Jay Molino, Asdrúal Rojas.

**Data curation:** Lina González, Issa Griffith, Alfredo Lescher, Jay Molino, Asdrúal Rojas.

**Formal analysis:** Lina González, Issa Griffith, Alfredo Lescher, Jay Molino, Asdrúal Rojas.

**Funding acquisition:** Lina González, Issa Griffith, Alfredo Lescher, Jay Molino, Asdrúal Rojas.

**Investigation:** Lina González, Issa Griffith, Alfredo Lescher, Asdrúal Rojas.

**Methodology:** Lina González, Issa Griffith, Alfredo Lescher, Jay Molino, Asdrúal Rojas.

**Project administration:** Lina González, Issa Griffith, Alfredo Lescher, Jay Molino, Asdrúal Rojas.

**Resources:** Lina González, Issa Griffith, Alfredo Lescher, Jay Molino.

**Software:** Lina González, Issa Griffith, Alfredo Lescher, Jay Molino, Damián Quijano.

**Supervision:** Lina González, Issa Griffith, Alfredo Lescher.

**Validation:** Lina González, Issa Griffith, Alfredo Lescher, Jay Molino, Damián Quijano.

**Visualization:** Alfredo Lescher, Jay Molino.

**Writing – original draft:** Jay Molino.

**Writing – review & editing:** Jay Molino, Asdrúal Rojas, Damián Quijano.

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
