## [Decision Letter · Decision Letter 0]

PONE-D-24-60715Evaluation of Flow Control Using PID versus Fuzzy Logic in an Electropneumatic Circuit for Pulmonary Ventilation ApplicationsPLOS ONE

Dear Dr. Molino,

Thank you for submitting your manuscript to PLOS ONE. After careful consideration, we feel that it has merit but does not fully meet PLOS ONE’s publication criteria as it currently stands. Therefore, we invite you to submit a revised version of the manuscript that addresses the points raised during the review process.

**ACADEMIC EDITOR: The manuscript has a potential. However, the authors have to carefully respond to the queries and suggestions raised by the reviewers. **

We look forward to receiving your revised manuscript.

Kind regards,

Shady H.E. Abdel Aleem, Phd

Academic Editor

PLOS ONE

“SENACYT Panama.”

“The authors would like to acknowledge the support of SENACYT via grant No. APY-NI2022-13.”

“SENACYT Panama.”

6. We note that your Data Availability Statement is currently as follows: [All relevant data are within the manuscript and its Supporting Information files.]

Additional Editor Comments:

The authors have to respond carefully to the queries and comments raised by the reviewers.

Reviewers' comments:

Reviewer's Responses to Questions

**Comments to the Author**

1. Is the manuscript technically sound, and do the data support the conclusions?

Reviewer #1: Yes

Reviewer #2: Yes

2. Has the statistical analysis been performed appropriately and rigorously? 

Reviewer #1: Yes

Reviewer #2: N/A

3. Have the authors made all data underlying the findings in their manuscript fully available?

Reviewer #1: Yes

Reviewer #2: Yes

4. Is the manuscript presented in an intelligible fashion and written in standard English?

Reviewer #1: Yes

Reviewer #2: No

5. Review Comments to the Author

Reviewer #1: -The introduction fails to clearly define the research gap. It is unclear why this specific comparison between PID and -Fuzzy Logic is necessary in the context of pulmonary ventilation.

-The literature review is outdated and does not adequately reference recent advancements in AI-based or adaptive control techniques for ventilation systems.

-The authors do not critically analyze the limitations of previous works; instead, they merely summarize existing studies without highlighting key shortcomings.

-The description of the electropneumatic circuit design is vague and lacks technical details. Important aspects such as system nonlinearities, sensor characteristics, and actuator dynamics are not well-explained.

-The fuzzy logic controller design lacks justification for the chosen membership functions and rule base. It is unclear why specific linguistic rules were selected and how they were optimized.

-The PID tuning parameters are not justified. The authors do not explain whether Ziegler-Nichols, trial-and-error, or any other systematic tuning method was used.

-No real-world validation: The study appears to be purely simulation-based without experimental verification on actual electropneumatic hardware. This severely limits the practical relevance of the results.

Reviewer #2: In this manuscript, the flow control is evaluated for a prototype electro-pneumatic unit for mechanical ventilator applications through the comparison between the testing results of PID and Fuzzy Logic method. Based on the testing data, the accuracy, the settling time and robustness are evaluated to illustrated the merits and the limits of the fuzzy logic and PID controllers. However, this manuscript contains too-many spelling and grammatical errors. The concrete comments are listed as follows.

1. Check for typos and grammatical mistakes throughout the manuscript to improve readability. For example: "Fuzzy Logic" and "fuzzy logic" should be unified, " quasi-experimental" is confused, "were taken" should be unified in the tense, “The main results show that according, the Bland-Altman analysis” is confused, “The purpose of the PID and Fuzzy Logic control in a prototype electro-pneumatic ventilator” is confused, "Equation 1" should be "Equation (1)" , etc.

2. The PID control algorithm is implemented using the Python library, however the determination of the constant parameters of Kp=5.7, Ki=87.3 and Kd=0.05 in this manuscript should be illustrated in detail.

3. Please illustrate the novelty of the control algorithm of PID and fuzzy logic method used in the tasting system, or the innovation of the experimental device and technique.

4. The parameters in Equations are confused, especially for “Z” and “z”, etc.

5. Unit should be added in the Figures and Tables.

6. The article layout is unreasonable. For example: “2.6 Ethical Statement” should not be in the body of the manuscript, “5. Authors contributions” should be after “8. Conclusion”, “6. Conflict of Interest Statement” and “7. Acknowledgments” should be deleted according to the guideline of author in PLOS One.

6. PLOS authors have the option to publish the peer review history of their article (what does this mean? ). If published, this will include your full peer review and any attached files.

**Do you want your identity to be public for this peer review?** For information about this choice, including consent withdrawal, please see our Privacy Policy .

Reviewer #1: **Yes: ** Dr. Gaurav Dhiman

Reviewer #2: No

---

## [Author Response · Author response to Decision Letter 1]

20 Apr 2025

Response to Reviewers Comments.

Ms. Ref. No.: PONE-D-24-60715

Title: Evaluation of Flow Control Using PID versus Fuzzy Logic in an Electropneumatic Circuit for Pulmonary Ventilation Applications.

We want to express our sincere gratitude to both reviewers for their thoughtful and constructive comments. Their evaluations were invaluable in helping us identify areas of improvement and refine the manuscript.

To provide precise and structured responses, we have categorized the comments thematically (e.g., Introduction, Methodology, Results, Language, and Format). This approach ensures clarity and alignment between the reviewers' observations and the corresponding revisions. Furthermore,we also improved the quality of the images.

We have addressed each concern carefully and revised the manuscript accordingly. We are confident that the updated version responds thoroughly to the reviewers' suggestions, both in content and in presentation.

Thank you once again for your contributions to strengthening the quality of our work.

Introduction

Reviewer #1

1. The introduction fails to clearly define the research gap. It is unclear why this specific comparison between PID and Fuzzy Logic is necessary in the context of pulmonary ventilation.

We appreciate the reviewer's suggestion and have significantly revised the introduction to define the research gap better. The revised introduction now explicitly states the limitations of conventional PID controllers in handling nonlinear respiratory dynamics, such as changing lung compliance and airflow resistance, which are common challenges in mechanical ventilation.

Furthermore, we highlight that while Fuzzy Logic Control (FLC) has been explored as an alternative, prior comparisons with PID often relied on theoretical models or simulations rather than direct experimental validation under identical test conditions. To address this gap, our study performs a real-world comparative evaluation of both controllers on an electropneumatic ventilator prototype, ensuring a fair and controlled assessment of their performance in regulating flow, handling disturbances, and adapting to variable lung mechanics.

Additionally, we now clarify this comparison's practical significance, particularly in ventilator shortages, where cost-effective, adaptive control strategies are crucial. The inclusion of recent studies on PID tuning, adaptive fuzzy controllers, and hybrid approaches (e.g., [8,10,12,14,15]) strengthens the motivation for this research by demonstrating that while advanced controllers exist, their comparative effectiveness in real ventilator systems remains an open question.

2. The literature review is outdated and does not adequately reference recent advancements in AI-based or adaptive control techniques for ventilation systems.

Thank you for this valuable feedback. We have updated the literature review in the introduction to incorporate recent advancements in AI-driven and adaptive ventilation control. The revised section now includes:

• Adaptive PID and model-based controllers: Studies on dynamic gain-tuning strategies for PID, such as Tyreus-Luyben tuning and auto-tuning algorithms ([17,18,19]).

• Fuzzy-based intelligent controllers: New research on hybrid fuzzy-PID systems and fuzzy inference applied to respiratory mechanics, highlighting their advantages in handling nonlinearities ([10,15,16,20]).

• Machine learning in ventilator control: Acknowledgment of data-driven approaches, including ANFIS (adaptive neuro-fuzzy inference systems) and AI-assisted tuning methods ([13,22,24]).

3. The authors do not critically analyze the limitations of previous works; instead, they merely summarize existing studies without highlighting key shortcomings.

We acknowledge this concern and have strengthened our critical discussion of previous studies in the introduction. The revised text now explicitly identifies three major shortcomings in prior PID vs. fuzzy logic comparisons:

• Lack of direct experimental validation: Many studies on fuzzy logic controllers rely on simulated patient models rather than real electro-pneumatic ventilator implementations, making it difficult to assess real-world applicability ([9,13,15]).

• Limited consideration of dynamic patient conditions: Traditional PID controllers are often evaluated under static lung compliance scenarios, whereas real patients exhibit significant variability that may degrade PID performance over time ([8,17,18]).

• Absence of comprehensive performance metrics: While prior research reports improvements in rise time and overshoot, few studies compare stability margins, steady-state error, and response to disturbances in a single experimental framework ([10,12,14]).

Our study directly addresses these gaps by conducting a controlled experimental comparison of both controllers on an electro-pneumatic ventilator, providing quantitative insights into their performance across multiple ventilation scenarios.

Methodology / Experimental Setup

Reviewer #1

1. The description of the electro-pneumatic circuit design is vague and lacks technical details. Important aspects such as system nonlinearities, sensor characteristics, and actuator dynamics are not well-explained.

We appreciate this observation. In the revised manuscript, we have significantly expanded the description of the electropneumatic system. Specific nonlinearities—such as valve hysteresis, sensor latency, and dynamic compliance/resistance of the test lung—are now explicitly addressed (Section: System Nonlinearities and Dynamic Characteristics). We also added detailed specifications for the sensors (Sensirion SFM3000) and actuators (Camozzi Series AP and CFB) in Table 1 and throughout the system description. This contextualization clarifies the sources of nonlinearity and their control implications.

2. The fuzzy logic controller design lacks justification for the chosen membership functions and rule base. It is unclear why specific linguistic rules were selected and how they were optimized.

We have expanded the explanation of the FLC design to include justification for both the membership functions and the rule base. Triangular membership functions were selected to ensure smooth transitions and computational efficiency. The 25-rule base was constructed to mimic expert respiratory therapy heuristics, using a 5×5 matrix mapping of flow error and error rate. Empirical tuning and system response analysis were employed to refine rule behavior and optimize stability, which is now explained in detail in the section Fuzzy Logic Controller (FLC) Design.

3. The PID tuning parameters are not justified. The authors do not explain whether Ziegler-Nichols, trial-and-error, or any other systematic tuning method was used.

We now clarify that the Ziegler-Nichols method was initially used to determine preliminary gain values (Kc and Tu), followed by empirical fine-tuning to mitigate overshoot and reduce steady-state error. This process is described under PID Controller Design and Tuning, along with the final tuning values (Kp=5.7, Ki=87.3, Kd=0.05). We also added commentary on why the initial Ziegler-Nichols estimates were not directly applied due to system sensitivity.

Reviewer #2

1. The PID control algorithm is implemented using the Python library; however, the determination of the constant parameters of Kp=5.7, Ki=87.3, and Kd=0.05 in this manuscript should be illustrated in detail.

As requested, we now provide a full description of the PID tuning process. The controller was initially tuned using the Ziegler-Nichols method to estimate critical gain and oscillation period, and the resulting gains were refined experimentally. The tuning process was guided by optimizing the response to step inputs, aiming to minimize overshoot and settling time while ensuring stability in the presence of nonlinear actuator and lung dynamics. This clarification is included in the PID Controller Design and Tuning section.

2. Please illustrate the novelty of the control algorithm of PID and fuzzy logic method used in the testing system, or the innovation of the experimental device and technique.

We appreciate the reviewer's observation regarding the novelty of our control algorithms and prototype design. In response, we have expanded the methodology section—particularly under "Experimental Validation"—to explicitly outline the innovative aspects of our system. The revised manuscript now emphasizes the following key contributions:

• Novel experimental setup: Unlike many previous studies that rely solely on simulation, our study presents a fully implemented electropneumatic ventilator prototype using commercial components (e.g., ESP32, Camozzi valves, Sensirion SFM3000 sensor). This allows for real-world evaluation of control strategies under dynamic respiratory conditions.

• Unique comparative framework: Both the PID and fuzzy logic controllers were implemented on the same physical prototype, allowing for a direct, controlled, and empirical comparison. This approach mitigates confounding factors often present in simulation-only or non-parallel studies.

• Controller implementation details: The PID controller was tuned using the Ziegler–Nichols method, followed by fine-tuning to address overshoot and delay under load. The fuzzy controller design uses two trapezoidal-input universes and a 12-rule base optimized for real-time response, implemented via Python and an LUT-based method to address memory limitations of the ESP32.

• Experimental rigor: The system's response was evaluated using step-response, perturbation rejection, and Bland–Altman analysis, offering a detailed empirical performance assessment rarely reported in similar ventilator controller studies.

• Low-cost and scalable design: The controller architecture, based on open-source hardware and libraries, supports replication and potential adaptation for emergency use in resource-constrained environments—a relevant innovation in the context of future pandemics or disaster-response medicine.

These additions are summarized in the newly added "Novelty of the Approach" paragraph within the Methodology section (see end of "Experimental Validation").

Results & Validation

Reviewer #1

1. No real-world validation: The study appears to be purely simulation-based without experimental verification on actual electropneumatic hardware. This severely limits the practical relevance of the results.

We thank the reviewer for this important comment. We respectfully clarify that the ventilator system described in this study is a fully constructed hardware prototype, not a simulation-only model. The experiments were conducted using physical components, including an ESP32 microcontroller, Camozzi Series AP and CFB valves, a Sensirion SFM3000 flow sensor, and a medical-grade Aridyne 3500 air compressor.

Validation was carried out using the ACCU LUNG test lung and VT650 flow and pressure analyzer, both manufactured by Fluke. These devices are industry-standard tools for verifying respiratory device performance under safe, repeatable, and clinically relevant simulated conditions. They enable the evaluation of control system behavior under variable compliance and resistance settings without the ethical and logistical complexities of patient testing.

We agree that clinical trials represent the next stage in ventilator development; however, our current objective is to provide a robust technical comparison of PID and Fuzzy Logic control strategies within a validated physical testbed. This approach aligns with standard ventilator development workflows and supports early-stage innovation, especially in the context of cost-effective, locally manufactured solutions.

Clarity & Language

Reviewer #2:

1. Check for typos and grammatical mistakes throughout the manuscript to improve readability. For example: 'Fuzzy Logic' and 'fuzzy logic' should be unified, 'quasi-experimental' is confused, 'were taken' should be unified in the tense, 'The main results show that according, the Bland-Altman analysis' is confused, 'The purpose of the PID and Fuzzy Logic control in a prototype electro-pneumatic ventilator' is confused, 'Equation 1' should be 'Equation (1)', etc.

We appreciate the reviewer's detailed feedback. A comprehensive grammatical and stylistic revision was performed throughout the manuscript. We believe we have addressed all the shortcomings in the style, specifically:

• Capitalization: Fuzzy Logic is consistently capitalized throughout the text to ensure uniformity and professionalism.

• Clarification of Study Design: The term quasi-experimental was removed from the Abstract to avoid ambiguity. It now reads:

A laboratory-based experimental study was conducted under laboratory conditions, using a test lung simulator and real-time flow data acquisition.

• Equation references were standardized in accordance with journal guidelines (e.g., "Equation (1)").

• Units have been added or clarified in figures and tables as needed.

2. The parameters in Equations are confused, especially for 'Z' and 'z', etc." "In Equations (2) and (3), the variable z denotes the complex frequency domain variable used in Z-transform analysis for discrete-time systems.

We thank the reviewer for this observation. To prevent confusion between the uses of the symbols 'Z' and 'z', we have clarified their meaning explicitly in the revised manuscript.

In the Repeatability section, the variable 'Z' refers to the critical value from the standard normal distribution used in the sample size estimation formula. We added the following sentence to clarify this usage:

Here, Z represents the critical value of the standard normal distribution corresponding to the desired confidence level.

In the Jury Analysis section, the variable 'z' denotes the Z-domain variable commonly used in transfer function representation for discrete-time systems. We clarified this by adding the sentence:

In Equations (2) and (3), the variable z denotes the complex frequency domain variable used in Z-transform analysis for discrete-time systems.

3. Unit should be added in the Figures and Tables.

We have revised all figures and tables to indicate appropriate units clearly.

• For example, in Table 2 (PID tuning parameters) and Table 4 (Fisher test values), all variables now include units where applicable (e.g., flow in L/min, pressure in psi or bar, time in seconds, etc.).

• In the Bland–Altman results (Table 2) and associated plots (Figure 2), units such as liters per minute (L/min) have been specified for clarity and consistency.

• Additionally, y-axis and x-axis labels and style in all plots and figures were reviewed and updated to reflect proper units and a homogeneus visuals.

Structure & Formatting Issues

Reviewer #2:

1. The article layout is unreasonable. For example: '2.6 Ethical Statement' should not be in the body of the manuscript, '5. Authors contributions' should be after '8. Conclusion', '6. Conflict of Interest Statement' and '7. Acknowledgments' should be deleted according to the guideline of author in PLOS One.

We thank the reviewer for the helpful observations regarding the manuscript structure. The article layout has been revised to comply with the journal's formatting guidelines. All required sections have been repositioned or removed as appropriate, ensuring consistency with PLOS ONE's submission standards. We appreciate the reviewer's input in helping us improve the clarity and organization of the manuscript.

---

## [Decision Letter · Decision Letter 1]

Evaluation of Flow Control Using PID versus Fuzzy Logic in an Electropneumatic Circuit for Pulmonary Ventilation Applications

PONE-D-24-60715R1

Dear Dr. Molino,

We’re pleased to inform you that your manuscript has been judged scientifically suitable for publication and will be formally accepted for publication once it meets all outstanding technical requirements.

Kind regards,

Shady H.E. Abdel Aleem, Phd

Academic Editor

PLOS ONE

Additional Editor Comments (optional):

Reviewers' comments:

Reviewer's Responses to Questions

**Comments to the Author**

1. If the authors have adequately addressed your comments raised in a previous round of review and you feel that this manuscript is now acceptable for publication, you may indicate that here to bypass the “Comments to the Author” section, enter your conflict of interest statement in the “Confidential to Editor” section, and submit your "Accept" recommendation.

Reviewer #1: (No Response)

Reviewer #2: All comments have been addressed

2. Is the manuscript technically sound, and do the data support the conclusions?

Reviewer #1: (No Response)

Reviewer #2: Yes

3. Has the statistical analysis been performed appropriately and rigorously? 

Reviewer #1: (No Response)

Reviewer #2: Yes

4. Have the authors made all data underlying the findings in their manuscript fully available?

Reviewer #1: (No Response)

Reviewer #2: Yes

5. Is the manuscript presented in an intelligible fashion and written in standard English?

Reviewer #1: (No Response)

Reviewer #2: Yes

6. Review Comments to the Author

Reviewer #1: I accept this work. I accept this work. I accept this work. I accept this work. I accept this work. I accept this work.

Reviewer #2: (No Response)

7. PLOS authors have the option to publish the peer review history of their article (what does this mean? ). If published, this will include your full peer review and any attached files.

**Do you want your identity to be public for this peer review?** For information about this choice, including consent withdrawal, please see our Privacy Policy .

Reviewer #1: No

Reviewer #2: No

---

## [Editor Report · Acceptance letter]

PONE-D-24-60715R1

PLOS ONE

Dear Dr. Molino,

I'm pleased to inform you that your manuscript has been deemed suitable for publication in PLOS ONE. Congratulations! Your manuscript is now being handed over to our production team.

Kind regards,

on behalf of

Professor Shady H.E. Abdel Aleem

Academic Editor

PLOS ONE